# Wild *Saccharomyces* Produced Differential Aromas of Fermented Sauvignon Blanc Must

**Sandra D. C. Mendes** [1,*], **Stefany Grützmann Arcari** [2], **Simone Silmara Werner** [1,3], **Patricia Valente** [4] and **Mauricio Ramirez-Castrillon** [5,*]

1    Agricultural Research and Rural Extension (Epagri), Experimental Station, Lages 88502-970, SC, Brazil; simoneswerner@gmail.com
2    Federal Institute of Santa Catarina, (IFSC), Campus São Miguel do Oeste, São Miguel do Oeste 89900-000, SC, Brazil; arcari.ste@gmail.com
3    Department of Informatics and Statistics, Universidade Federal de Santa Catarina, Florianópolis 88040-370, SC, Brazil
4    Department of Microbiology, Immunology and Parasitology, Institute of Basic Health Sciences, Universidade Federal do Rio Grande do Sul, Porto Alegre 90050-170, RS, Brazil; patricia.valente@ufrgs.br
5    Research Group in Mycology, Faculty of Basic Sciences, Universidad Santiago de Cali, Cali 760035, Colombia
*    Correspondence: mendes@epagri.sc.gov.br (S.D.C.M.); mauriciogeteg@gmail.com (M.R.-C.)

**Abstract:** Nine *Saccharomyces* strains, previously isolated from vineyards in Southern Brazil, were used as starter cultures in fermentations of Sauvignon Blanc (SB) must at laboratory scale, to study inter-strain differences in aroma profiles. The molecular profiles differentiated the following isolates from the reference strain (SC2048), which is typically used in wine production: 06CE, 11CE, 33CE, 01PP, 12M, 13PP, 26PP, 28AD, and 41PP. Under the same conditions, each of these strains produced different concentrations and combinations of metabolites, which significantly influenced the aroma of the fermented SB must. Volatile compounds such as octanoic acid, diethyl succinate, and ethyl lactate were associated with the strains 26PP, 41PP, 01PP, and 12M, while strains 33CE, 28AD, 13PP, and 06CE were associated with the production of ethyl acetate and 1-hexanol. Strain 06CE produced 592.87 ± 12.35 µg/L 1-hexanol. In addition, the olfactory activity values (OAVs; we considered only values >1) allowed us to evaluate the participation of each compound in the aroma of the final fermented SB. In conclusion, the selected wild strains are promising candidates for improving the regional characteristics of wine.

**Keywords:** aroma profile; volatile compounds; fermentations; yeast; metabolites

## 1. Introduction

Various civilizations have sought to control the processes of fermenting beverages and food by selecting yeasts that are specifically adapted to their needs. Over the centuries, the use and reuse of specific yeasts have produced strains that resulted from this selection process. Domesticated strains that have been selected for certain phenotypic traits are expected to be genetically differentiated [1]. Strains used for wine fermentation have reduced levels of genetic variation, and this is the result of a genetic bottleneck produced by the selection for specific characteristics. These characteristics include the fermentation of sugars present in the must (<2.0 g/L residual sugar), low $H_2S$ production, tolerance to osmotic and ethanol stress, tolerance to temperature variation and $SO_2$, and low foam production [2,3]. Higher alcohols are formed during the metabolism of sugars or amino acids through catabolic and anabolic pathways [4,5]. Volatile fatty acids are synthesized in different yeast metabolism pathways depending on the chain length and branching type [5], while the formation of esters is the result of the interaction of alcohols and byproducts of yeast fermentation [5].

According to metagenomics studies, the diversity of industrial yeasts represents a small fraction of that among existing species [6–8]. The importance of this standardization

ensures control and wine quality inside a winery [9]. However, it reduces the possibilities to offer unique wines highlighting the added value of a regional characteristic [10]. In this sense, the particularities of wine currently depend only on the grape and geographical region, and not the aroma offered by autochthonous yeasts.

"Vinhos de Altitude" is one of the vineyards grown in Santa Catarina, Brazil, which obtained geographical indication in 2021 [11]. The quality of wines produced in this region is recognized by the soil characteristics, altitude, grape varieties, and culture techniques. Sauvignon Blanc (SB) cultivars from this region are cultured between 947 and 1415 m of altitude, with soluble solids between 17.8° and 20° Brix, and a pH of 2.94–3.32. SB wines contain 12.2–12.4% *v/v* of ethanol and are described as wines with high aromatic complexity [10]. We hypothesized that the aroma profile of Sauvignon Blanc wines, produced in the traditional "Vinhos de Altitude" region, could be differentiated by the contribution of aromas provided by wild yeast originating from this region. Thus, we aimed to assess the flavor profiles of wild strains of *Saccharomyces cerevisiae* in fermented Sauvignon Blanc musts at laboratory scale. We studied the chemical composition of the unfermented and fermented musts. These chemicals, consisting mainly of ethyl esters, higher alcohols, and fatty acids, are of great relevance in determining the quality of wine.

## 2. Materials and Methods

### 2.1. Microorganisms

Nine strains of *Saccharomyces* were used in this study; they were isolated from leaves, bunches of grapes, and soil from vineyards located at "Planalto Serrano", Santa Catarina, Brazil. This region has a geographical indication, named "Vinhos de altitude" in the São Joaquim region (28°16′30′′ S, 49°56′09′′ W, alt 1400 m). Strain enrichment and isolation were performed as previously described by Mendes et al. [12], and we used a commercial strain, *Saccharomyces cerevisiae* SC2048 (CLIB), as reference. All stock cultures were stored at $-30\,^{\circ}$C in 20% *v/v* glycerol and grown in complete YPD broth (5 g/L yeast extract, 20 g/L peptone, 20 g/L dextrose) for 72 h at 30 $^{\circ}$C. Strains were identified on the basis of sequences from two regions: the D1/D2 domain of the large ribosomal subunit (LSU), and the ITS1–5.8S–ITS2 region [13]. In addition, a species-specific primer set (ScerF2 and ScerR2) was used to identify the *Saccharomyces* species, as described by Muir et al. [14]. The yeast strains were discriminated using the Intron Splice Site primer EI-1 (5′–CTGGCTTGGTGTATGT–3′) [15].

### 2.2. Fermentation Experiments

Sauvignon Blanc (SB) grapes were grown in São Joaquim, Santa Catarina, Brazil. The SB extract was treated with 29.0 g/L of sulfur dioxide in the form of sodium metabisulfite ($Na_2S_2O_5$), 8.0 mL/L of pectolytic enzyme to clarify the must, and 7.0 mL/L of bentonite (bentogran®, AEB, San Polo, Italy) to facilitate the sedimentation of nonfermentable solids. The unfermented must was then transferred to the cold storage chamber (4 $^{\circ}$C) and incubated for 7 days. After clarification, a total volume of 200 mL of the unfermented SB must was filter-sterilized through a nitrocellulose membrane (0.22 μm pore size, 47 mm in diameter). Filtered and unfermented SB must (4.9 mL) was homogenized and aseptically transferred to 20 mL vials, ensuring the same chemical composition of the must for all experiments, such as sugar concentration (22.2 g/L) or pH (3.5). On average, the YAN concentration for SB musts obtained from "Vinhos de altitude" is 246.54 mg N/L [16]. All experiments were performed at laboratory scale. Each of the nine wild strains and the reference were cultured at 28 $^{\circ}$C for 24 h in YPD broth. The cell density of the exponentially growing yeast cells was adjusted to $2.0 \times 10^7$ cells/mL, and then 100 μL of each strain was inoculated into each vial containing 4.9 mL of unfermented SB must. Finally, the unfermented must was incubated at 25 $^{\circ}$C for 48 h.

### 2.3. Preparation of SPME Fibers

SPME fibers made of DVB/CAR/PDMS (Supelco; Bellefonte, PA, USA) were conditioned according to the manufacturer's recommendations. Then, we added 1.5 g NaCl to

each vial containing 4.9 mL of the fermented SB must. The solution was incubated for 5 min at 56 °C. We then exposed the SPME fiber to the headspace (HS) for 55 min, which was then analyzed by gas chromatography (GC-FID, Thermo Scientific Trace 1310, Waltham, MA, USA). The desorption in the gas chromatograph injector was performed for 2 min at a temperature of 265 °C in splitless mode.

*2.4. Qualitative Analysis of Unfermented SB Must*

The volatile compounds in unfermented SB must were identified using methods described by Arcari et al. [17]. A Varian CP-3900 (USA) gas chromatograph equipped with a Varian Saturn 4000 trap ion mass spectrometer (GC-MS), and the Saturn GC-IT/MS version 5.51 Workstation software were used to identify volatile compounds. Compounds were separated using a ZB-WAXplus (60 m × 0.25 mm × 0.25 µm) column from Zebron (USA), with helium gas as the carrier gas at a flow rate of 1.0 mL·min$^{-1}$. The temperature program was an initial oven temperature of 40 °C for 5 min increasing by 2 °C per min until reaching 220 °C. The ion trap detector was operated at temperatures of 200 °C in the transfer line, 50 °C in the manifold, and 180 °C in the trap. All mass spectra were obtained by electron impact (70 eV) in scan mode (25–400 $m/z$). The emission current was 50 µA, with a maximum ionization of 25,000 µs. The positive identification of the compounds was performed by comparison of the retention time obtained for the sample with that observed for the standards of the volatile compounds injected under the same conditions and based on a comparison of the mass spectra with those given in the spectral database of the National Institute of Standards and Technology (NIST) MS 05, considering above 80% similarity. MS data processing also utilized the automated mass spectral deconvolution and identification system (AMDIS) program version 2.71. The retention index (LTPRI—linear temperature-programmed retention index) was also calculated using a commercial hydrocarbon mixture (C$_8$–C$_{20}$). The tentative identification of other volatile compounds present in the sample was performed by comparing the LTPRI and the mass spectra obtained for the sample with the LTPRI reported in the literature and the mass spectra in NIST.

*2.5. Quantitative Analysis of Fermented SB*

The following volatile compounds were purchased from Sigma-Aldrich (Saint Luis, EUA): ethyl acetate (141-78-6), ethyl butanoate (105-54-4), ethyl pentanoate (539-82-2), ethyl hexanoate (123-66-0), ethyl heptanoate (106-30-9), ethyl octanoate (106-32-1), ethyl nonanoate (123-29-5), ethyl ecanoate (110-38-3), ethyl undecanoate (627-90-7), ethyl dodecanoate (106-33-2), diethyl succinate (123,25-1), ethyl lactate (97-64-3), ethyl cinnamate (103-36-6), ethyl anthranilate (87-25-2), ethyl isobutanoate (97-62-1), ethyl 3-hydroxybutanoate (5405-41-4), ethyl isovalerate (108-64-5), ethyl 2-methylbutanoate (7452-79-1), phenylethyl acetate (103-45-70), hexyl acetate (142-92-7), *S*-furfuryl thioacetate (13678-68-7), furfuryl acetate (623-17-6), isobutyl acetate (110-19-0), isoamyl acetate (123-92-2), 3-methyl-1-butanol (123-51-3), methanol (67-56-1), 1-butanol (71-36-3), 2-butanol (78-92-2), 1-propanol (71-23-8), 2-phenylethanol (60-12-8), 1-hexanol (111-27-3), furfuryl alcohol (98-00-0), propanoic acid (79-09-4), butanoic acid (107-92-6), valeric acid (109-52-4), hexanoic acid (142-62-1), heptanoic acid (111-14-8), octanoic acid (124-07-2), pelargonic acid (112-05-0), decanoic acid (334-48-5), undecanoic acid (112-37-8), 10-undecenoic acid (112-38-9), isobutyric acid (79-31-2), isovaleric acid (503-74-2), α-pinene (7785-70-8), β-pinene (19902-08-0), geraniol (106-24-1), α-terpineol (98-55-5), limonene (5989-27-5), citronellal (2385-77-5), cedrene (469-61-4), γ-nonalactone (104-61-0), β-damascenone (23696-85-7), α-ionone (127-41-3), β-ionone (14901-07-6), and 4-methyl-2-pentanol (108-11-2).

Volatiles were quantified as described recently by Arcari et al. [17] using a Thermo Scientific Trace 1310 (Waltham, MA, USA) gas chromatograph equipped with a flame ionization detector (GC-FID) and ChromQuest software (Waltham, MA, USA). The contribution of a chemical compound to the overall aroma of fermented SB was quantified through the olfactory activity value (OAV). The OAV is an indicator of the importance



of a specific compound in the aroma of the sample and was calculated as the ratio of the concentration of the individual compound relative to the threshold of perception described in the literature. Only those compounds that reached OAV >1 were considered important for the overall aroma of SB. The clustering of OAV for each strain was represented by a heat map as described by [18].

### 2.6. Statistical Analysis

All assays were carried out in triplicate. All data are presented as means, along with their standard deviations, and coefficients of variation. ANOVA and a post hoc Tukey HSD were performed to evaluate differences on aroma profiles for yeast strains. *p*-Values ≤0.05 were considered statistically significant. Principal component analysis (PCA) was used to determine the best aroma profile and to discriminate between aroma profiles of the different strains. All statistical analyses were performed with owner scripts and the packages openxlsx [19], gplots [20], and FactoMineR [21] based on the R Core Team [18].

## 3. Results

### 3.1. Selection and Identification of Saccharomyces Strains

The isolates, collected from different areas within the vineyards, were identified by sequencing the D1/D2 domain of the large ribosomal subunit (LSU) or the ITS1–5.8S–ITS2 region (ITS). We used the species-specific primers ScerF2 and ScerR2 to identify *S. cerevisiae* species and to distinguish them from other *Saccharomyces* species that can be found in fermented musts, namely, *S. bayanus*, *S. pastorianus*, and *S. kudriavzevii* [14]. Of the nine *Saccharomyces* isolates, six were identified as *S. cerevisiae*, according to amplicon lengths of 150 bp. Meanwhile, PCR assays using the Intron Splice Site primer EI-1 on strains 06CE, 11CE, and 33CE showed a different profile from the rest of the vineyard isolates. Strains 06CE, 11CE, and 33CE did not assimilate α-trehalose, maltose, and raffinose; the ability to assimilate these sugars was a strain-specific characteristic. Based on our results, we suggest that these strains showed a different physiological and genetic profile from other ones.

### 3.2. Determination of Volatile Compounds before Fermentation

A total of 44 volatile compounds were identified in the unfermented SB must (Table 1). From them, the identities of 20 volatile compounds were verified using commercial standards. Unidentified volatile compounds were compared to the similarities between the mass of the sample compounds and those of the NIST library (similarity > 70%). In addition, we also compared the calculated retention index to those of polar columns of polyethylene glycol found in the literature. Similarly, we observed deviations of at least 29 units for 3-hexen-2-one and isocitronellol. The volatile fraction of unfermented SB must was composed mostly of esters (12 compounds), followed by higher alcohols (11), acids (five), ketones (four), C13-norisoprenoids (four), aldehydes (three), terpenes (three), phenol (one), and lactone (one). The compounds with higher concentrations were hexanoic acid (19.24 ± 4.85 μg/L), octanoic acid (19.60 ± 0.02 μg/L), and decanoic acid (10.67 μg/L), suggesting that they are not noticeable by humans according to the perception threshold [22], while other volatile compounds, such as ethyl isobutanoate (50.24 ± 0.46 μg/L) and ethyl pentanoate (15.95 ± 0.68 μg/L) could be noticeable [23] (Table 1). We highlight the presence of β-damascenone and β-ionone as compounds that were characteristic for the unfermented must, but with a low presence in fermented musts.

### 3.3. Determination of Volatile Compounds after Fermentation

Grape-derived compounds such as terpenes, pyrazines, and thiols play a key role in the aroma of SB, while alcoholic fermentation by *S. cerevisiae* induces the formation of active secondary aroma metabolites such as ethyl esters, higher alcohols, and fatty acids, as demonstrated by PCA. The first two principal components resulting from comparing chemical groups in the volatile fraction explained 71.51% of the total variance of the data

(Figure 1a). Chemical groups such as terpenes, fatty acids, ethyl esters, and lactones had a positive effect on the first PC (Figure 1a), while acetate esters and higher alcohols had positive effects on the second PC (Figure 1c). Figure 1a shows that strains 26PP, 12M, 41PP, and 01PP had significant positive factor loadings in component 1 and were associated with the production of terpenes, fatty acids, and ethyl esters, while strains 06CE and 13PP had significant positive factor loadings in component 2 and were associated with the production of higher alcohols and acetate esters. Strain 06CE was associated with the production of 1-hexanol (592.87 ± 12.35 µg/L), ethyl acetate (7574.84 ± 1786.28 µg/L), and furfuryl acetate (25.54 ± 4.93 µg/L, Figures 1c and 2). The unique strain that produced heptanoic acid was 26PP (63.07 ± 1.98 µg/L). Considering the ethyl ester and acetate ester groups, strains 41PP and 26PP presented loadings for component 1, while 13PP presented positive loadings for component 2, standing out from the others (Figure 1b). Strain 12M presented positive loadings for component 1 and 2, with a different pattern compared to all other strains, yielding 193.64 ± 92.04 µg/L ethyl heptanoate (Figure 1b). Strains 11CE, 28AD, and 33CE and the unfermented SB must had significant negative factor loadings in component 1 when considering higher alcohols, $C_{13}$-norisoprenoids, and fatty acids (Figure 1c, Tables 2 and 3).

**Table 1.** Volatile compounds in unfermented Sauvignon Blanc (SB) must, along with their respective retention times, identification ions, retention indices, identification methods, odor descriptor, and perception threshold.

| Retention Time | Compound | Selected Ions | Estimated LTPRI [**] | LTPRI from Literature | Identification Method | Odor Descriptor | Perception Threshold (µg L$^{-1}$) |
|---|---|---|---|---|---|---|---|
| 7592 | Ethyl acetate | 61, 88 | 891 | 890 [e] | STD [***], MS [****] | Solvent [a,b], fruity [c,d], balsamic [d] | 12,000 [e] |
| 10,937 | Ethyl isobutanoate | 116, 88, 71 | 966 | 968 [e] | STD, MS | Fruity, banana [h] | 15 [e] |
| 17,674 | Ethyl pentanoate | 88, 57, 85 | 1126 | 1132 [e] | STD, MS | Fruity, apple [e] | 5 [e] |
| 18,193 | 1-Butanol | 39, 57, 72 | 1159 | 1165 [e] | MS | Medicinal [e] | 150,000 [e] |
| 18,262 | Thioacetic acid | 43,61, 42 | 1167 | 1163 [j] | MS | Toasted [j], onion, garlic [j] | Nf |
| 19,012 | 3-Hexen-2-one | 83, 55, 43 | 1182 | 1211 [n] | MS | Boiled vegetables, metal [j] | Nf |
| 21,544 | 2 6-Dimethyl-4-heptanone (isovalerone) | 57, 85, 41 | 1195 | Nf | MS | Nf [*] | Nf |
| 24,320 | 3-Methyl-1-butanol | 42, 55, 70 | 1202 | 1205 [j] | STD, MS | Burnt, alcohol [c,h], nail polish, whiskey [d] | 30,000 [l] |
| 25,217 | 2-Hexanol | 45, 69, 41 | 1217 | 1238 [n] | MS | Nf | Nf |
| 34,006 | 1-Hexanol | 56, 69, 84 | 1369 | 1372 [l] | STD, MS | Herbaceous, greasy [i], resinous; floral, green, cut grass [d,h] | 110 [e] |
| 34,592 | (E)-3-Hexen-1-ol | 41, 67 | 1376 | 1379 [n] | MS | Herbaceous [j] | 70 [o] |
| 35,861 | (Z)-3-Hexen-1-ol | 67, 41 | 1394 | 1401 [n] | MS | Herbaceous, bitter, fatty [e] | 1000 [e] |
| 36,653 | 2,4-Hexadienal | 81, 39, 41 | 1402 | 1407 [n] | MS | Vegetable [j] | 60 [o] |
| 37,351 | Isocitronellol | 83, 55, 41 | 1459 | 1488 [j] | STD, MS | Candy, roses [j] | 40 [o] |
| 41,250 | Linalool oxide | 59, 43 | 1480 | 1484 [n] | MS | Candy, floral, woody [j] | 500 [e] |
| 43,139 | Benzaldehyde | 51, 77, 106 | 1493 | 1529 [e] | MS | Almonds [e] | 2000 [e] |
| 43,587 | Isovaleric acid | 60, 43, 41 | 1666 | 1660 [e] | STD, MS | Candy, cheese [k], rancidity [e] | 3000 [e] |
| 43,921 | Ethyl decanoate | 88, 101, 29 | 1668 | 1651 [e] | STD, MS | Fruity, grape [e] | 200 [l] |
| 44,040 | Diethyl succinate | 101, 129, 29 | 1691 | 1690 [e] | STD, MS | Wine [c,d,h], toffee [f], fruity [d] | 200,000 [l] |
| 44,734 | Acetophenone | 105, 77, 51 | 1692 | 1690 [j] | MS | Floral, almonds [j] | 65 [o] |
| 44,946 | α-Terpineol | 81, 136, 43 | 1711 | 1713 [e] | STD, MS | Floral, candy [e], anise, mint [j] | 250 [l] |
| 44,959 | 1,1,6-Trimethyl-1,2-dihydronaphthalene (TDN) | 142, 159, 172 | 1697 | 1714 [n] | MS | Liqueur [n] | 540 [m] |
| 45,552 | 1-Decanol | 70, 55, 56 | 1722 | 1735 [n] | MS | Candy, fatty [e] | 400 [e] |
| 46,065 | Verbenone | 107, 91, 39 | 1725 | 1742 [n] | MS | Mint, spices [n] | Nf |

| Retention Time | Compound | Selected Ions | Estimated LTPRI ** | LTPRI from Literature | Identification Method | Odor Descriptor | Perception Threshold (µg L$^{-1}$) |
|---|---|---|---|---|---|---|---|
| 47,829 | 1-Undecanol | 55, 69, 41 | 1737 | 1738 [e] | MS | Fruity [e], tangerine [j] | 41 [e] |
| 61,241 | α-Ionone | 136, 121, 93 | 1808 | 1829 [n] | STD, MS | Fruity, floral, raspberry, violet [h] | 2,6 [l] |
| 61,925 | β-Damascenone | 190, 121, 69 | 1815 | 1842 [e] | STD, MS | Baked apple [l], floral, honey [d,l] | 0,05 [l] |
| 62,388 | Ethyl laurate | 88, 101 | 1838 | 1856 [n] | STD, MS | Candy, floral [e], waxy, soap [h] | 1500 [e] |
| 63,044 | Hexanoic acid | 60, 73, 41 | 1869 | 1863 [e] | STD, MS | Cheese, greasy [e] | 420 [l] |
| 63,759 | Decyl isobutyrate | 43, 89, 71 | 1870 | Nf | MS | Nf | Nf |
| 64,386 | Benzyl alcohol | 79, 108, 107 | 1871 | 1874 [n] | MS | Candy, fruity [e] | 200,000 [l] |
| 66,097 | 2-Phenylethanol | 65, 91, 92 | 1931 | 1939 [n] | STD, MS | Roses, honey [e,k] | 14,000 [l] |
| 70,677 | Phenol | 94, 66, 65 | 1968 | 1962 [n] | MS | Phenolic, medicinal [n] | 5900 [o] |
| 71,382 | β-Ionone | 177, 192, 91 | 1985 | 1975 [n] | STD, MS | Violet [d,h,i], balsamic, roses [d] | 0,09 [l] |
| 72,105 | Isopropyl myristate | 43, 102, 60 | 1999 | 2017 [n] | MS | Nf | 800 [e] |
| 72,695 | Ethyl myristate | 88, 101, 43 | 2025 | 2044 [n] | MS | Lily [j] | Nf |
| 72,881 | γ-Nonalactone | 85, 29, 41 | 2032 | 2044 [n] | STD, MS | Coconut, peach [b,g,j] | 30 [l] |
| 73,384 | Octanoic acid | 60, 73, 43 | 2048 | 2055 [n] | STD, MS | Rancidity [d,k], candy, cheese [c], animal, spices [f], unpleasant [d] | 500 [l] |
| 80,756 | Ethyl cinnamate | 103, 131, 176 | 2140 | 2139 [j] | STD, MS | Honey, cinnamon [c,f], floral, strawberry, plum [f] | 1,1 [l] |
| 82,127 | Ethyl palmitate | 88, 101 | 2234 | 2250 [j] | MS | Waxy, greasy [e] | 1500 [e] |
| 83,100 | Decanoic acid | 60, 129, 172 | 2279 | 2287 [e] | STD, MS | Unpleasant [d,k], rancid fat [c], animal [f] | 1000 [e] |
| 83,273 | Ethyl-9-hexadecenoate | 55, 88, 69 | 2279 | 2265 [j] | MS | Nf | Nf |
| 87,629 | 2-Hexadecanol | 55, 69, 83 | 2310 | 2302 [e] | MS | Nf | Nf |
| 96,623 | Hexyl cinnamaldehyde | 129, 117, 91 | 2512 | 2526 [j] | MS | Nf | Nf |

* Nf = not found. ** LTPRI = linear temperature-programmed retention indexes. *** STD = mass spectra and retention index in agreement with the standard of the volatile compound. **** MS = mass spectra in agreement with the spectral database (NIST considering minimum 70% similarity). [24] [a], [25] [b], [26] [c], [27] [d], [23] [e], [28] [f], [29] [g], [30] [h], [31] [i], [17] [j], [32] [k], [22] [l], [33] [m], [34] [n], [35] [o].

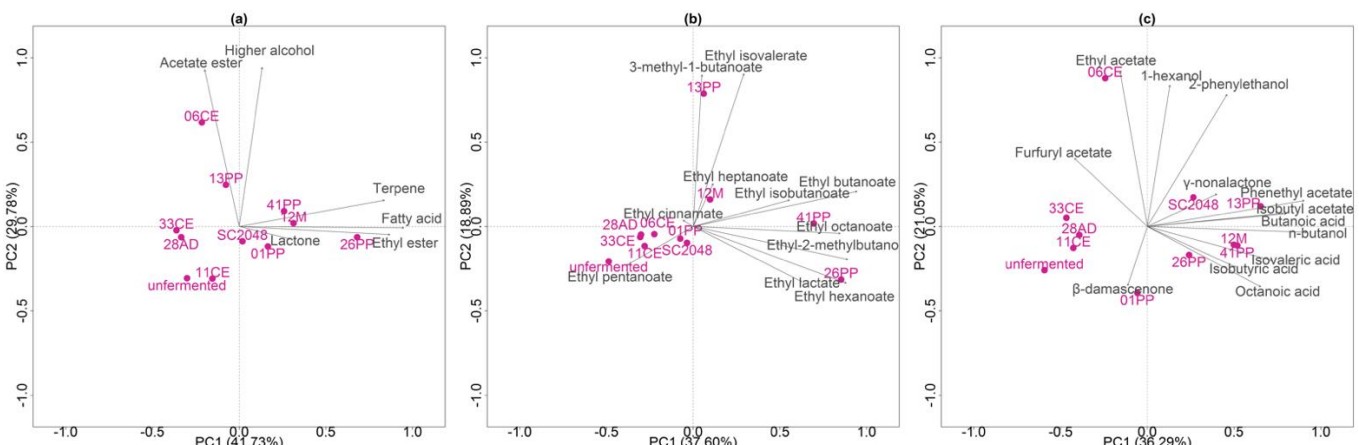

**Figure 1.** Principal component analysis (PCA) of volatile compounds produced in unfermented and fermented Sauvignon Blanc (SB) musts. (**a**), Inoculation with a commercial strain of *S. cerevisiae* (2048SC), wild strains of *Saccharomyces* spp. (26PP, 41PP, 01PP, 12M, 33CE, 28AD, 13PP, 06CE, and 11CE), and unfermented SB must (**b**). Principal component analysis of ethyl ester groups. (**c**), Principal component analysis of higher alcohols, C$_{13}$-norisoprenoids, and fatty acids.

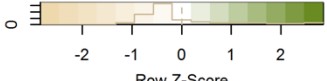

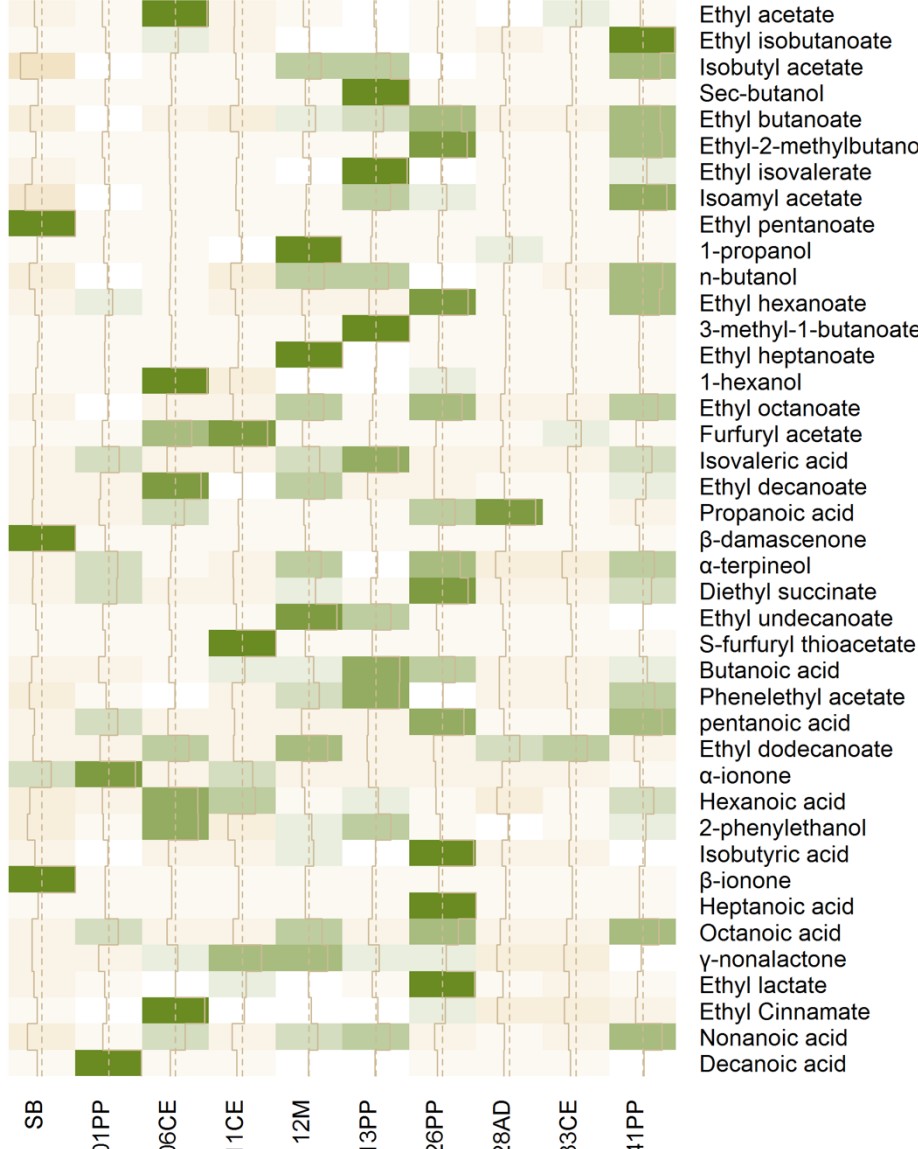

**Figure 2.** Heatmap generated from the *z*-scores representing the olfactory activity values of the volatile compounds produced by wild strains 26PP, 41PP, 01PP, 12M, 33CE, 28AD, 13PP, 06 EC, and 11CE and controls (unfermented SB must and commercial strain 2048SC). The values in green and yellow represent high and low olfactory activities, respectively; white represents no change in the aroma perception of the volatile compounds.

**Table 2.** Concentration of fatty acids (µg/L) in unfermented and fermented Sauvignon Blanc musts. Means ± standard error, followed by the same letter, do not differ according to Tukey's test considering a 0.05 significance level.

| Sample | Isovaleric Acid | Propanonic Acid | Butanoic Acid | Pentanoic Acid | Hexanoic Acid | Isobutyric Acid | Octanoic Acid | Nonanoic Acid | Decanoic Acid |
|---|---|---|---|---|---|---|---|---|---|
| Unfermented SB must | 3.15 ± 0.00 ** b | nd | nd | nd | 19.24 ± 4.85 cd | nd | 19.60 ± 0.02 d | nd | 10.67 ± 0.00 c |
| 01PP | 418.07 ± 7.40 a | nd | nd | 115.95 ± 15.67 a | 204.49 ± 20.83 abc | 2469.61 ± 942.96 b | 35,294.01 ± 7207.02 bc | 22.68 ± 3.11 cd | 313.82 ± 103.16 a |
| 06CE | nd | 61.95 ± 25.65 a | 22.9 ± 0.97 c | nd | 514.82 ± 201.34 a | nd | 1939.97 ± 20.6 cd | 124.65 ± 0.98 ab | 7.26 ± 0.28 abc |
| 11CE | 3.15 ± 0.00 b | 18.18 ± 1.16 b | 78.25 ± 21.28 bc | nd | 383.73 ± 7.76 ab | nd | 656.85 ± 194.51c | 20.00 ± 1.75 d | 2.25 ± 0.26 c |
| 12M | 423.14 ± 41.13 a | 15.27 ± 1.57 b | 90.44 ± 26.58 bc | nd | 126.68 ± 38.38 abcd | 4668.82 ± 1139.40 b | 41,355.92 ± 7212.70 a | 60.64 ± 6.12 cd | 39.26 ± 0.03 ab |
| 13PP | 689.03 ± 75.02 a | 15.87 ± 0.63 b | 211.96 ± 15.52 a | nd | 225.58 ± 109.20 abc | 2709.97 ± 694,67 b | 4516.95 ± 437.15 cd | 74.55 ± 11.62 bc | 38.35 ± 13.68 ab |
| 26PP | nd | 74.53 ± 1.51 a | 141.72 ± 3.39 abc | 201.68 ± 18.64 a | 93.67 ± 6.18 abcd | 1,501,940 ± 442.93 a | 47,158.69 ± 1563.37 a | 17.23 ± 9.32 d | 3.59 ± 0.49 bc |
| 28AD | nd | 121.81 ± 22.46 a | nd | 14.66 ± 0.11 a | 7.24 ± 0.79 d | nd | 1144.48 ± 59.19 cd | 25.07 ± 11.36 cd | 0.33 ± 0.12 d |
| 33CE | nd | 11.25 ± 2.28 bc | nd | 15.48 ± 2.91 a | 100.40 ± 5.09 bcd | nd | 533.61 ± 220.57 cd | 18.19 ± 0.00 d | 0.30 ± 0.07 d |
| 41PP | 387.82 ± 53.47 a | 7.31 ± 0.32 c | 91.48 ± 1.03 bc | 187.10 ± 44.34 a | 351.77 ± 14.98 ab | 3554.93 ± 163.85 b | 47,297.21 ± 9350.84 a | 87.31 ± 5.72 bc | 53.92 ± 5.71 a |
| SC2048 | nd | nd | 170.37 ± 24.46 ab | 68.03 ± 6.28 a | 121.15 ± 17.15 bcd | 2439.60 ± 40.93 b | 17,624.72 ± 1071.05 ab | 130.01 ± 16.83 a | 51.87 ± 0.46 a |
| Mean * | 174.94 ± 53.15 | 29.65 ± 8.51 | 73.37 ± 16.12 | 54.81 ± 16.61 | 195.34 ± 36.87 | 2805.66 ± 920.47 | 17,958.36 ± 4365.11 | 52.76 ± 9.62 | 47.42 ± 20.11 |
| *p*-Value | <0.0001 | <0.0001 | 0.0050 | 0.3165 | 0.0013 | 0.0002 | <0.0001 | <0.0001 | <0.0001 |

* Obtained considering the nd samples as zero. ** Very similar concentrations between replicates, with very low standard deviation trending to zero.

**Table 3.** Concentration of ethyl esters (µg/L) in unfermented and fermented Sauvignon Blanc musts. Means ± standard error, followed by the same letter, do not differ according to Tukey's test considering a 0.05 significance level.

| Sample | Ethyl Isobutanoate | Ethyl Butanoate | Ethyl 2-Methylbutanoate | Ethyl Isovalerate | Ethyl Hexanoate | Ethyl Heptanoate | Ethyl Octanoate | Ethyl Decanoate | Diethyl Succinate | Ethyl Undecanoate | Ethyl Laurate | Ethyl Lactate | Ethyl Cinnamate |
|---|---|---|---|---|---|---|---|---|---|---|---|---|---|
| Unfermented SB must | 50.24 ± 0.46 b | nd | nd | nd | nd | nd | nd | nd | 9.95 ± 0.39 c | nd | nd | nd | 4.36 ± 0.00 a |
| 01PP | 62.81 ± 2.80 b | 31.69 ± 4.39 cd | nd | 65.25 ± 11.62 b | 54.14 ± 21.52 ab | nd | 299.63 ± 31.31 a | nd | 13,394.20 ± 1138.25 a | nd | nd | 3662.88 ± 841.64 a | 3.05 ± 1.31 a |
| 06CE | 114.53 ± 12.92 b | 14.98 ± 0.72 ef | nd | 50.97 ± 15.81 b | 4.37 ± 0.03 b | 0.24 ± 0.02 c | 10.00 ± 1.52 a | 7.42 ± 2.65 ab | 1600.93 ± 540.20 bc | 1.06 ± 0.15 bc | 7.58 ± 0.71 ab | 7691.16 ± 333.83 a | 13.01 ± 2.91 a |
| 11CE | 20.28 ± 0.28 b | nd | 9.31 ± 0.00 ** c | 49.40 ± 13.04 b | 0.11 ± 0.00 c | 0.29 ± 0.28 b | 0.14 ± 0.00 b | 1.83 ± 0.83 abc | 400.81 ± 107.15 c | 0.21 ± 0.15 bc | nd | 11,490.42 ± 3910.71 a | 4.36 ± 0.00 a |
| 12M | 92.07 ± 33.33 ab | 53.06 ± 8.13 bc | nd | 134.30 ± 76.67 b | nd | 193.64 ± 92.04 a | 865.29 ± 464.55 a | 5.22 ± 1.41 a | 11,968.00 ± 4026.20 ab | 3.23 ± 0.68 a | 9.30 ± 0.20 a | 7368.24 ± 170.89 a | 4.36 ± 0.00 a |
| 13PP | 97.12 ± 13.90 ab | 57.44 ± 1.40 abc | 12.83 ± 0.57 b | 464.22 ± 87.49 a | nd | 24.50 ± 4.43 a | 214.34 ± 79.12 a | nd | 3261.78 ± 114.68 abc | 2.00 ± 0.10 ab | nd | 4461.40 ± 215.54 a | 4.35 ± 0.01 a |
| 26PP | 46.12 ± 12.5 b | 90.10 ± 3.30 a | 111.13 ± 12.97 a | 96.87 ± 44.16 ab | 162.88 ± 27.39 a | nd | 1025.26 ± 376.89 a | nd | 26,709.35 ± 712.21 a | nd | nd | 39,063.35 ± 8671.66 a | 4.57 ± 0.46 a |

**Table 3.** *Cont.*

| Sample | Ethyl Isobutanoate | Ethyl Butanoate | Ethyl 2-Methylbutanoate | Ethyl Isovalerate | Ethyl Hexanoate | Ethyl Heptanoate | Ethyl Octanoate | Ethyl Decanoate | Diethyl Succinate | Ethyl Undecanoate | Ethyl Laurate | Ethyl Lactate | Ethyl Cinnamate |
|---|---|---|---|---|---|---|---|---|---|---|---|---|---|
| 28AD | 30.59 ± 0.46 b | 7.88 ± 1.36 f | nd | 68.07 ± 10.39 ab | 5.74 ± 2.60 b | 2.58 ± 0.05 a | 15.47 ± 0.36 a | 0.74 ± 0.11 bcd | 393.49 ± 53.44 c | nd | 6.49 ± 0.00 b | 1618.32 ± 198.99 a | nd |
| 33CE | 38.11 ± 16.82 b | 7.58 ± 1.57 f | nd | 51.94 ± 2.41 b | 6.87 ± 0.04 b | nd | 0.185 ± 0.06 b | 0.58 ± 0.10 cd | 352.81 ± 119.04 c | 0.09 ± 0.03 c | 6.79 ± 0.23 ab | 1138.68 ± 25.63 a | nd |
| 41PP | 360.28 ± 5.63 a | 85.22 ± 10.20 ab | 91.90 ± 2.70 a | 157.16 ± 27.02 ab | 117.28 ± 15.11 ab | nd | 873.51 ± 272.29 a | 2.51 ± 0.66 ab | 13,859.32 ± 1514.52 a | 0.60 ± 0.19 bc | nd | 2382.97 ± 539.76 a | 1.43 ± 0.00a |
| SC2048 | 60.87 ± 3.54 b | 23.51 ± 2.49 de | nd | 74.49 ± 14.21 b | 23.76 ± 1.67 b | nd | 526.43 ± 12.23 a | 0.46 ± 0.04 d | 7186.51 ± 1142.79 abc | nd | nd | 14,727.44 ± 1852.39 a | nd |
| Mean * | 88.45 ± 19.94 | 33.77 ± 6.92 | 20.47 ± 8.48 | 110.24 ± 27.38 | 34.10 ± 11.87 | 20.11 ± 13.51 | 348.20 ± 94.93 | 1.71 ± 0.55 | 7194.30 ± 1797.07 | 0.65 ± 0.23 | 2.74 ± 0.81 | 8509.53 ± 2399.36 | 3.59 ± 0.80 |
| *p*-Value | 0.0036 | <0.0001 | <0.0001 | 0.0258 | <0.0001 | <0.0001 | <0.0001 | 0.0025 | 0.0001 | 0.0041 | 0.0309 | 0.1668 | 0.6426 |

* Obtained considering the nd samples as zero. ** Very similar concentrations between replicates, with very low standard deviation trending to zero.

Figure 2 shows the compounds that can be perceived by the human nose, grouped into three clusters. Furthermore, it shows the relative levels of volatile compounds produced by each strain and the importance of the compounds in differentiating their profile from that of the commercial strain, SC2048. The higher alcohols that stood out were 1-hexanol, 2-phenylethanol, and 1-butanol, while the fatty acid esters that stood out were ethyl-2-methyl butanoate, ethyl butanoate, ethyl cinnamate, ethyl heptanoate, ethyl isobutanoate, and ethyl isovalerate. Among acetate esters, only phenylethyl acetate was present at levels higher than 1.0 for the OAV. Among $C_{13}$-norisoprenoids, $\alpha$-ionone, $\beta$-damascenone, and $\gamma$-nonalactone stood out; these compounds, even at low concentrations, contributed positively to SB flavor.

## 4. Discussion

The molecular identification of yeasts permitted a reduction in the number of strains for assessment of fermentation experiments. The main barcodes for fungi, ITS and LSU regions, generally show low variability in the genus *Saccharomyces*, hindering the distinction of *S. cerevisiae* from other species. Among nine strains previously identified as *Saccharomyces* by sequencing the LSU and/or ITS regions, only three strains were not identified as *S. cerevisiae* according to multiplex PCR using the specific primers ScerF2 and SceR2. Due to differences in the physiological tests, genetic profiles, and source of isolation, we suggest that strains 06CE, 11CE, and 33CE are hybrids of *S. cerevisiae* and other *Saccharomyces* species. In fact, natural environments and industrial fermentations could have led to spontaneous formation of interspecific hybrids between *S. cerevisiae, S. kudriavzevii, S. uvarum,* and *S. eubayanus* [14,36,37]. These strains could be different in copy number, ploidy variations, genome rearrangements, and polymorphism changes [38]. After strain identification, the Intron Splice Site primer EI-1 was used to detect polymorphisms between the commercial yeast (SC2048) and wild strains 33CE, 11CE, and 06CE, suggesting that they should be different to *S. cerevisiae*. However, we cannot confirm with our results that they were interspecific hybrids.

The chemical composition of grapes strongly influences the formation of aroma compounds by *Saccharomyces* species. In many cases, the yeast releases an aromatic compound from a nonvolatile precursor molecule. The genetic profile of *S. cerevisiae* is relevant in the formation of metabolites that confer specific aromas after fermentation and, consequently, in wine. Moreover, several other factors also affect the spectrum of aroma compounds formed. Because of this, we characterized the unfermented SB must (Table 1) before fermenting it with the selected strains. C6 alcohols (1-hexanol, 2-hexanol, (*E*)-3-hexen-1-ol, and (*Z*)-3-hexen-1-ol) were present in the must, as a product of enzyme activity on linoleic and linolenic acids extracted from the grapes during the crushing stage. 1-Hexanol is one of the main compounds associated with the aroma of Sauvignon Blanc wines from Victoria, Australia, and Marlborough, New Zealand [39,40]. Among the strains assessed in this study, the most notable was the strain 06CE, whose production of 1-hexanol was identified as a positive contributor. Its presence is associated with nuances of herbal and resinous aromas, as well as the sensory characteristic of cut grass wine; therefore, it has been identified as an important compound [41]. The 12M strain showed concentrations of 1-propanol that can significantly influence the sensory properties of the wine. In fact, the fruity aroma at total concentrations below 300 mg/L of higher alcohols could contribute positively to the aromatic profile of wines, increasing fruity and flowery notes and aroma complexity. However, at levels above 400 mg/L, there should appear a negative effect caused by the apparition of pungent and unpleasant notes [42,43]. Meanwhile, strain 01PP was the major producer of both $\beta$-damascenone and $\alpha$-ionone, showing the highest intensity for red berry aromas, followed by floral aromas. Their sensory detection thresholds are 50 ng/L and 90 ng/L in hydroalcoholic solution, respectively, indicating their potential importance to wine flavor [44]. Wines of monoculture have much higher OAVs of fruity, floral, and sweety profiles, where *S. cerevisiae* strains are protagonists due to the production of higher amounts of ethyl octanoate, ethyl decanoate, $\beta$-damascenone, and phenylacetaldehyde. In compar-

ison, chemical, fatty, and herbaceous aroma series are equal in monoculture wines [45]. The accumulation of compounds in must is related to the anabolism of grape precursors, which stimulate the production of volatile compounds by yeast. The strain 26PP showed a different pattern than all other strains, producing about $63.07 \pm 1.98$ μg/L heptanoic acid. This group of compounds, essentially esters, alcohols, and acids (mainly C4–C10 fatty acids), may provide an important link between grape composition and the volatile profile of the fermented must [46–48]. Thus, the overall aroma of wine is influenced by complex interactions between several components and is seldom dominated by a single component. These observations are consistent with those of Knight et al. [7] and Escudero et al. [49], who detected a significant effect of selected strains of *S. cerevisiae* on grape phenotypes in each region.

## 5. Conclusions

In this study at laboratory scale, the strains 01PP, 06CE, 12M, 41PP, and 26PP showed traits suggesting that they may be promising candidates as inoculants (such as commercial yeast). These selected strains belong to the *Saccharomyces* genus, and they modified the aroma profile of fermented Sauvignon Blanc must, characterized by fruity and flowery notes. Specific yeast strains produced highlighted ethyl esters, such as ethyl isovalerate (13PP), ethyl acetate (06CE), ethyl hexanoate (26PP), ethyl heptanoate (12M), and ethyl octanoate (41PP). In addition, there were marked differences between strains in terms of their associated aromas, as indicated by their OAVs. We suggest assessing the potential of these yeast strains in co-fermentations on a pilot scale (micro-vinification). Furthermore, we suggest performing a sensorial analysis during the winemaking process and aging for Sauvignon Blanc wine at "Vinhos de altitude", in Brazil.

**Author Contributions:** Conceptualization, S.D.C.M. and M.R.-C.; data curation, S.S.W.; formal analysis, S.S.W.; funding acquisition, S.D.C.M. and M.R.-C.; methodology, S.D.C.M. and S.G.A.; supervision, P.V.; validation, S.D.C.M. and S.G.A.; visualization, S.S.W.; writing—original draft, S.D.C.M. and M.R.-C.; writing—review and editing, S.G.A., P.V., and M.R.-C. All authors have read and agreed to the published version of the manuscript.

**Funding:** This research was funded by EMBRAPA, grant number "seplan 6313858". The APC of this research was funded by Dirección General de Investigaciones at Universidad Santiago de Cali, under call No. 01-2022.

**Institutional Review Board Statement:** Not applicable.

**Informed Consent Statement:** Not applicable.

**Data Availability Statement:** The sequencing data associated with this study were deposited in GenBank under accession number PRJNA532626.

**Acknowledgments:** The authors are grateful to the winery/vineyard owners for allowing us to collect the samples; financial support was provided by Epagri/Ufrgs. The graphical abstract was created with Biorender.com.

**Conflicts of Interest:** The authors declare no conflict of interest. The funders had no role in the design of the study; in the collection, analyses, or interpretation of data; in the writing of the manuscript, or in the decision to publish the results.

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
