# Peer review of "Wild Saccharomyces Produced Differential Aromas of Fermented Sauvignon Blanc Must"

_fermentation, doi:10.3390/fermentation8040177_

Round 1
Reviewer 1 Report
The authors presented a good manuscript focused on the fermentative performance of indigenous strains on the flavor profile of Sauvignon Blanc. In my opinion, it is appropriate to implement the introduction section that is too skimpy;
Why didn't the authors add an analysis of the basic chemical parameters?
Section 3.2 regarding the presentation of results should be expanded, with a more detailed presentation of the results;
In Tables 1 and 2 there are standard deviations (sd) equal to "0.00", not correct, please correct;
Author Response
Dear reviewer. We thank you in advance for each comment, that enriched our manuscript. We attached a tracked version of the manuscript to follow the changes made during revision. In the next lines, we replied to each comment.
The authors presented a good manuscript focused on the fermentative performance of indigenous strains on the flavor profile of Sauvignon Blanc. In my opinion, it is appropriate to implement the introduction section that is too skimpy;
R// Thank you very much for your comments. We added more information in the introduction about the particularities of “Vinhos de Altitude” that we believed should enrich the information about the location of the experiments. Also, we added information about the effect of yeasts on wine aroma, during fermentation. Please, see the modified text inside the section Introduction.
Why didn't the authors add an analysis of the basic chemical parameters?
R// Physicochemical analyzes are commonly performed on wines to establish the oenological parameters that classify them according to international oenological practices (OIV). Since we studied samples of fermented and unfermented must, but not wine, we only performed analyzes for must characterization, such as pH and sugar concentration. Furthermore, the study aimed to evaluate the aroma profile of fermented Sauvignon blanc must, to establish the contribution of wild yeasts isolated from the same geographic region, to produce volatile compounds responsible for the secondary aroma (from fermentation). As the vinification process was not carried out in its entirety (it was not our objective to study the maturation of the fermented product), the oenological parameters would not result in significant variables for discussion in this study.
Section 3.2 regarding the presentation of results should be expanded, with a more detailed presentation of the results;
R// We transferred the Table S1 to the main text in the manuscript as Table 1. We considered that these results of qualitative identification should expand the view of volatile compounds that could be not quantified but detected by Mass Spectrometry. Also, we modified this section to highlight main volatile compounds detected by this method.
In Tables 1 and 2 there are standard deviations (sd) equal to "0.00", not correct, please correct;
R// We verified the original data, and the absolute values were identical between replicates. For this reason, the standard deviations in some cases were trending to 0.00. We included as a footnote in each table a note highlighting that this standard deviation is the result of three measurements for each variable, and the absolute value was very similar.

Reviewer 2 Report
The authors analyse the aroma production of different Saccharomyces strains in the grape variety Sauvignon Blanc. They perform adequat statistical analysis of their data and give some outlines for further experiments.
The introduction could describe more in detail the impact of yeast's aroma contribution.
Sensory analysis could be performed in order to confirm the chemical data.
Author Response
Dear reviewer. We thank you in advance for each comment, that enriched our manuscript. We attached a tracked version of the manuscript to follow the changes made during revision. In the next lines, we replied to each comment.
The authors analyse the aroma production of different Saccharomyces strains in the grape variety Sauvignon Blanc. They perform adequat statistical analysis of their data and give some outlines for further experiments. The introduction could describe more in detail the impact of yeast's aroma contribution.
R// Thank you very much for your comments. We added more information in the introduction about the particularities of “Vinhos de Altitude” that we believed should enrich the information about the location of the experiments. Also, we added information about the effect of yeasts on wine aroma, during fermentation.
Sensory analysis could be performed in order to confirm the chemical data.
R// Sensory analysis is only feasible in wine samples. Considering that the objective of this study was to determine the profile of volatile compounds responsible for the aroma of unfermented and fermented Sauvignon blanc must, to establish the contribution of wild yeasts to the production of secondary aroma, , we did not perform sensory analysis. Therefore, aroma changes were analyzed by GC-MS and GC-FID (Arcari et al., 2017), all well-established methods for aroma characterization in must samples.
Considering that only around 10% of volatile compounds have an important contribution to the final aroma of the wine, the calculation of the odor activity value (OAV) is important for the study of chemical compounds capable of stimulating a sensory response by the human olfactory system. The olfactory impact of each volatile compound is dependent on its concentration and its threshold of perception by the human nose, which varies considerably for each compound (Jiang et al., 2013; Guth, 1997; Vilanova; Martínez, 2007; Smith; Cozzolino, 2013). Therefore, when it is not possible to perform sensory analysis, the OAV calculation is important to estimate the contribution of volatile compounds to the aroma.

Round 2
Reviewer 1 Report
I accept this version of the manuscript
This manuscript is a resubmission of an earlier submission. The following is a list of the peer review reports and author responses from that submission.
Round 1
Reviewer 1 Report
Please, see comments attached

Author Response
Dear reviewer, we thank you for all your comments and recommendations. In the next lines, we replied point by point in each paragraph.
Manuscript 1577426 is about the analysis of fermented/unfermented Sauvignon Blanc using a number of yeast strains isolated from vines and soil. This analysis included secondary aromas (alcoholic fermentation by-products) such as esters, and primary volatiles such as C13-norisoprenoids and monoterpenes. Although the research appears to have been conducted with great care, I believe there are significant gaps in the study, which prevents the manuscript to be published at this stage, at least in my opinion.
First and foremost, an investigation of varietal thiols, which are the most relevant chemicals for S. blanc (4-MMP, 3-MH, 3-MHA) MUST have been included. I'm not clear why they looked at C13- norisoprenoinds and terpenes (both of which are generated from grapes) but not thiols (and eventually methoxypyrazines).
R//
Dear reviewer, we appreciate all your comments. Our focus in this research was to assess compounds produced directly by yeast metabolism. Specifically, we looked for esters that affect the wine aroma after fermentation. Many thiols are formed depending on sulfur, and several authors showed that thiols are strongly dependent on fermentation temperature, but not by the yeast strain (Masneuf-Pomarède et al. 2006, Duc et al. 2020). In fact, they are not produced by yeast metabolism, showing that the effect of thiols in Sauvignon blanc are dependent on the must (Duc et al. 2020). Thiols are involved in many oxidative and hydrolytic reactions that occur during wine aging, which are mainly affected by temperature (Waterhouse et al., 2016). However, post-fermentative aromas, or bouquet scents, that are produced during aging, were not assessed in this work because our objective was only to evaluate the effect of yeasts in fermentations at laboratory scale (Arcari et al , 2017; SWIEGERS et al., 2005). Therefore, we prioritized the study of sulfur compounds such as S-furfuryl thioacetate and furfuryl acetate, since yeast strain can have a large impact on the biosynthesis of these sulfur compounds.
Regarding methoxypyrazines, it is well known that the concentration of these compounds are dependent on the grape genotype and, during alcoholic fermentation, they are readly extracted from the skins. Considering that the concentration of methoxypyrazines is not affected by the yeast strain used in the fermentation, our study chose not to present the concentration of these compounds. On the other hand, C-13 norisoprenoids and monoterpenes were included in this study, especially β-damascenone. High concentrations of isoprenoids have been reported for musts and wines from the geographical region of the unfermented Sauvignon blanc must used in this study (Falcão et al., 2007; Falcão et al., 2008).
The timeliness of the analysis is the second important drawback of this issue. SPME-GC/MS analysis should have been performed on finished/bottled wine (after at least 8-9 months from the end of fermentation). I'm not sure when this analysis was performed, but it looks that no winemaking/aging was done once the fermentation was completed.
R//
Considering that this is a study at laboratory scale, with the aim of evaluating the effect of nine yeast strains, selected from the terroir of grape production, on the formation of volatile compounds, the vinification process was not carried out completely. The results obtained show that some studied yeast strains have potential for use in co-fermentations, in order to improve the increment of esters. However, the effect of using these strains on wine production, including the entire winemaking process, will be studied in future work.
We modified the title of the manuscript:
“Differing aromas of fermented “Sauvignon blanc” must pro-duced from wild Saccharomyces”.
Also, we modified the conclusions:
“In this study at laboratory scale, the strains 01PP, 06CE, 12M, 41PP and 26PP showed traits that suggest that they may be promising candidates as inoculants (like commercial yeast). These selected strains belonged to the Saccharomyces genus and modified the aroma profile of fermented Sauvignon Blanc must, characterized by fruit and flowery notes. In addition, there were marked differences between strains in terms of their associated aromas, as indicated by their OAVs. We suggest assessing the potential of these yeast strains in co-fermentations in a pilot scale (microvinification). Also, we suggest performing a sensorial analysis during winemaking process and aging for Sauvignon Blanc wine at “Vinhos de altitude”, in Brazil.”
A third key pitfall is a lack of knowledge on how the YAN was managed throughout fermentation. It is generally known that YAN has an effect on esters, among other classes of volatiles. However, there is no information available. The authors should have set the YAN of each ferment to 220-240 mg/L
R//
We appreciate the important note regarding YAN content. Knowing that the concentration of YAN influences the kinetics of fermentation and the final sugar concentration in the fermented product, the Sauvignon blanc must was standardized for all fermentation tests. Therefore, the concentration of YAN, fermentable sugars, organic acids etc. were the same for all treatments studied.
Minor issues:
• Figures 1/2: labels are unreadable
R//
To better understanding of the manuscript, we combined Figures 1 and 2, and we increased the size of the words. Also, we changed the word “SB” by “unfermented” to highlight the unfermented SB must.
• Always specify ‘unfermented’ or ‘fermented’ must
R//
Corrected
• L67-73: replace ‘mixture’ with unfermented must
R//
Corrected
• OAV is an important parameter but does not really give certainty of what each compound with give in terms in aroma. This is why a sensory analysis is often necessary to confirm this sort of results. The authors should include this in their conclusions.
R//
We agree with the reviewer in the sense that a sensory analysis will confirm these results. However, as the objective of our study was to analyze the aroma profile with fermented musts, i.e. without stabilization, bottling, aging, etc, it was not possible to perform this analysis. We decided to include an OAV analysis to approach the effect of each compound in the aroma. We included a recommendation in the conclusions about the possiblity to perform sensorial analysis in future works.
• B-damascenone >20 µg/L in a S. blanc appears to be very high. Please, double check its peak
R//
We verified the peak of β-damascenone and its concentration is correct. High concentrations of β-damascenone were previously reported in wines from the geographical region of the studied Sauvignon blanc must (Falcão et al., 2008).
• Tables with concentrations should have been included in the text (at least those from the SPME-GC.MS with the concentrations).
R//
The Tables S2 and S3 were moved to main text in the manuscript as Tables 1 and 2.
Reviewer 2 Report
The authors presented an interesting study on the fermentation performance of selected strains on the aroma of Sauvignon Blanc. However, there are major revisions needed to improve the manuscript.
Introduction: the introduction is scarce; the authors should go into more detail on the problems related to the use of selected strains, better define the aim of the work and the characteristics of the cv examined.
2.3 Preparation of SPME fibers
What sample volume was used?
2.4 Qualitative analysis of SB must
Explain the GC-MS conditions of separation and identification of the analytes
Results
The authors performed a qualitative analysis of the analytes in the must, and quantitative analysis of the analytes after fermentation. This creates a lot of confusion. It would have been more useful to also quantify the analytes in the must in order to discuss the evolution of volatile molecules during fermentation. Why was this not done?
Table S2: Replace the comma with a full stop.
Conclusions
Better define the results of the work
Author Response
Dear reviewer, we thank you for all your comments and recommendations. In the next lines, we replied point by point in each paragraph.
The authors presented an interesting study on the fermentation performance of selected strains on the aroma of Sauvignon Blanc. However, there are major revisions needed to improve the manuscript.
Introduction: the introduction is scarce; the authors should go into more detail on the problems related to the use of selected strains, better define the aim of the work and the characteristics of the cv examined.
R//
We thank you for all your comments and suggestions. We modified the introduction to highlight the problem of strain selection in the winemaking process. We also modified the objective of the work. The modified text is:
Line 39:
“Selected and domesticated yeast strains generally are used as starter cultures in the winemaking process, standardizing the aroma profile in the final product, wine. Therefore, the particularities of the wine are depending mainly on the grape and geo-graphical region. We hypothesized that the aroma profile of Sauvignon Blanc wines, produced in the traditional “Vinhos de Altitude” region, could be differentiated by contribution of aromas provided by wild yeast originating of this region. Thus, we aimed to assess the flavor profiles of wild strains of Saccharomyces cerevisiae in fermented “Sauvignon blanc” musts, obtained from vineyards with geographical indication “Vinhos de Altitude” in Santa Catarina, Brazil [7]. We studied the chemical composition of the unfermented and fermented musts. These chemicals, consisting mainly of ethyl esters, higher alcohols, and fatty acids, are of great relevance in determining the quality of wine.”
2.3 Preparation of SPME fibers
What sample volume was used?
R//
It was used 4.9mL by sample. We added this volume to the manuscript.
2.4 Qualitative analysis of SB must
Explain the GC-MS conditions of separation and identification of the analytes
R//
We added the conditions in the methodology (section 2.4).
Results
The authors performed a qualitative analysis of the analytes in the must, and quantitative analysis of the analytes after fermentation. This creates a lot of confusion. It would have been more useful to also quantify the analytes in the must in order to discuss the evolution of volatile molecules during fermentation. Why was this not done?
R//
Actually, the analysis was done before and after fermentation. Sorry by the confusion. We modified the figures to highlight the position of “unfermented Sauvignon blanc” must in the PCA analysis and Tables 1 and 2.
Table S2: Replace the comma with a full stop.
R//
Corrected.
Conclusions
Better define the results of the work
R//
We modified the conclusions
“​​In this study at laboratory scale, the strains 01PP, 06CE, 12M, 41PP and 26PP showed traits that suggest that they may be promising candidates as inoculants (like commercial yeast). These selected strains belonged to the Saccharomyces genus and modified the aroma profile of fermented Sauvignon Blanc must, characterized by fruit and flowery notes. Specific yeast strains produced highlighted ethyl esters, such as ethyl isovalerate (13PP), ethyl acetate (06CE), ethyl hexanoate (26PP), ethyl heptanoate (12M), and ethyl octanoate (41PP). In addition, there were marked differences between strains in terms of their as-sociated aromas, as indicated by their OAVs. We suggest assessing the potential of these yeast strains in co-fermentations in a pilot scale (microvinification). Also, we suggest performing a sensorial analysis during winemaking process and aging for Sauvignon Blanc wine at “Vinhos de altitude”, in Brazil.”
Round 2
Reviewer 1 Report
Dear Authors
Thank you for your response; nevertheless, I believe the paper is reporting on a research that is in the very early stages of being published, at least in my perspective. Despite the fact that the text has improved, I still feel that doing a volatile analysis on fermented juice rather than finished wine adds nothing to the field, especially given the volatiles' quick changes over time.
There's nothing wrong with completing a study on simply fermented juice, but to truly grasp the influence of these yeasts, a thorough investigation should have been done after at least 6-7 months. Furthermore, wines should have been treated like commercial wines, and therefore stabilised, and so on.
The authors also indicated that YAN (which has a significant influence on esters) was consistent throughout all trials, but the value is not provided in the text.
Reviewer 2 Report
Dear Authors,
standard deviations need to be added in tables 1 and 2.